# Assessment of the Impact of Chronic Pain on the Prevalence of Depressive Disorders in Patients with Endometriosis

**DOI:** 10.3390/diseases13090291

**Published:** 2025-09-02

**Authors:** Edyta Rysiak, Anna Grajewska, Anna Łońska, Jakub Tomaszewski, Karolina Kymona, Joanna Rostkowska

**Affiliations:** 1Department of Medicinal Chemistry, Faculty of Pharmacy, Medical University of Bialystok, 15-089 Białystok, Poland; edyta.rysiak@umb.edu.pl (E.R.);; 2Tomaszewski Medical Center of Obstetrics and Gynecology, Elizy Orzeszkowej 17/47, 15-084 Białystok, Poland; 3Department of Deaf Education and Research in the Education and Rehabilitation of Individuals with Hearing Impairments, Institute of Special Education, The Maria Grzegorzewska University, Szczęśliwicka 40, 02-353 Warszawa, Poland

**Keywords:** endometriosis, chronic pain, depression, complexity of care

## Abstract

Background: Endometriosis is a chronic, estrogen-dependent inflammatory and immunological disease, with chronic pain being its predominant clinical manifestation. This condition significantly impairs quality of life and is frequently associated with depressive and anxiety symptoms, further exacerbating social and occupational dysfunction in affected women. The aim of this study was to assess the relationship between chronic pain in patients with endometriosis and the severity of depressive symptoms. Methods: A retrospective analysis was conducted on the medical records of 60 women of reproductive age treated at the Tomaszewski Medical Center in Białystok between 2023 and 2024. Pain intensity was evaluated using the Visual Analogue Scale (VAS) and the McGill Pain Questionnaire, while depressive symptoms were assessed with the Beck Depression Inventory (BDI). Results: Statistical analyses included the Student *t*-test, Wilcoxon signed-rank test, chi-square test, and Shapiro–Wilk test, with significance set at *p* < 0.05. Pain intensity was significantly higher during menstruation (M = 7.23) compared to non-menstrual phases of the cycle (M = 4.55; *p* < 0.001). Accompanying symptoms included sleep disturbances, reduced activity, and gastrointestinal complaints. Depressive symptoms were also more severe during menstruation (M = 30.12) than during the rest of the cycle (M = 22.15; *p* < 0.001). A significant association between pain severity and depressive symptoms was observed during menstruation (χ^2^(4) = 12.89; *p* = 0.012), but not outside this phase. Conclusions: (1) Pain in endometriosis is chronic and cyclic in nature. (2) Depressive symptoms are common but may be masked by nonspecific somatic complaints. (3) Pain intensity strongly correlates with the severity of depressive disorders, particularly during menstruation. (4) The coexistence of depression significantly impairs patient functioning. (5) Effective management of endometriosis should integrate gynecological treatment with psychological support and psychiatric care when necessary.

## 1. Introduction

Endometriosis is a chronic, estrogen-dependent disorder characterized by the presence of endometrial-like tissue outside the uterine cavity. It is estimated to affect 10–15% of women of reproductive age, up to 50% of those with infertility, and more than half of patients reporting chronic pelvic pain [1,2]. Despite its high prevalence, diagnosis remains challenging due to the heterogeneity of symptoms and their overlap with conditions such as irritable bowel syndrome and inflammatory bowel disease. As a result, women often face a diagnostic delay of 4–11 years, with an average of approximately seven years before a definitive diagnosis is established [3]. The etiopathogenesis of endometriosis is multifactorial, involving both genetic and environmental determinants. Heritability is estimated to account for nearly 50% of disease risk, while epigenetic modifications triggered by environmental exposures are believed to facilitate progression from subtle lesions to clinically significant disease [4]. Clinically, the most burdensome manifestations include chronic pelvic pain, dyspareunia, pain during defecation, and infertility [5]. Pain, reported by the majority of affected individuals, is typically chronic and cyclical, profoundly impairing quality of life, occupational performance, and social functioning [6,7]. Importantly, the perception of pain in endometriosis extends beyond nociceptive mechanisms, encompassing cognitive and emotional dimensions that shape individual pain experience [8]. Functional pain in endometriosis is not necessarily attributable to tissue damage but may result from altered pain signal processing within the nervous system, further underscoring its multidimensional character) [9]. An increasing body of evidence highlights the psychological burden associated with endometriosis. The prevalence of depressive symptoms among patients is reported to range from 9.8% to 98.5%, and anxiety from 11.5% to 87.5%, rates that are substantially higher than those observed in women without the disease [10]. Women experiencing chronic endometriosis-related pain, in particular, exhibit significantly higher levels of depression compared to those with asymptomatic lesions or to healthy controls [11,12]. These findings indicate that pain is not only a central clinical symptom but also a critical factor in shaping the psychological profile of affected women. Nevertheless, the literature to date has largely emphasized either the biological underpinnings of endometriosis or general descriptions of psychiatric comorbidities, without adequately addressing the specific relationship between pain severity and depressive symptomatology. Clarifying this association is of clinical importance, as it may inform the development of integrated therapeutic approaches that address both the gynecological and psychological dimensions of the disease.

It is hypothesized that higher levels of pain experienced by women with endometriosis are positively correlated with the severity of depressive symptoms, a relationship grounded in theoretical models of chronic pain and substantiated by prior empirical findings. The aim of the present study is therefore to examine whether greater pain intensity in women with endometriosis is associated with more severe depressive symptoms.

## 2. Materials and Methods

The aim of the present study was to assess the relationship between the presence of chronic pain in patients diagnosed with endometriosis and the occurrence of emotional and depressive disorders. Considering the complexity of endometriosis as a research topic, the following research hypotheses were formulated:Among patients with endometriosis, the presence of chronic pain can be identified.Among patients with endometriosis, depressive disorders with a distinct clinical profile can be observed.There is a correlation between the intensity of pain perception and the emotional state of patients.The coexistence of depressive disorders has a significant impact on the functioning of patients with endometriosis.

### 2.1. Participant Characteristics

A retrospective analysis was conducted on medical records of 60 patients treated at the Tomaszewski Medical Center Clinic in Białystok between 2023 and 2024. The study included women of reproductive age diagnosed with chronic pain associated with endometriosis, who were not diagnosed with autoimmune diseases and were undergoing conservative treatment with hormonal pharmacotherapy. The research was based on the transcription of data extracted from the medical records, specifically from the Interview Scenario, which were interpreted at the time of analysis. The data concerned the following variables:Age,Socio-social conditions of the patients’ daily functioning,Co-occurrence of symptoms and ailments negatively affecting patients’ functioning,Pain-related complaints reported by the patients,Symptoms of a depressive nature.

### 2.2. Materials and Research Tools

Data were collected at the Tomaszewski Medical Centre Clinic by a physician and a psychologist. The forms included the Visual Analogue Scale (VAS), the McGill Pain Questionnaire, and the Beck Depression Inventory. Each patient responded to the questions included in the provided forms, which served as the interview framework. Prior to further analysis, all records were anonymised to eliminate any information that could allow for patient identification. The anonymised data were entered into an Excel spreadsheet, where individual entries were organised into cells to create a structured dataset. Data accuracy was verified using the Excel “data validation” function, which restricted the type of data permitted in each cell and ensured both consistency and correctness of the entered values.

The acquired data were categorized as follows:Assessment of pain intensityVisual Analogue Scale (VAS) [13]

The Visual Analogue Scale (VAS) is a tool used for the subjective assessment of the intensity of pain experienced by the patient. It enables a quantitative evaluation of pain severity. The patient’s current pain sensation is compared to the most intense pain they can imagine. The patient indicates the perceived pain level on a graphical scale spanning 10 cm in length, with the beginning of the line defined as “no pain” (0), and the end representing the “worst imaginable pain” (10). Figure 1. presents the graphical form of the scale.

This scale allows the definition of pain intensity by selecting the description that best matches the pain experienced:0—no pain1–3—mild pain4–6—moderate pain7–9—severe pain10—extreme (unbearable) pain

3.McGill Pain Questionnaire [14]

The McGill Pain Questionnaire consists of categories of words describing the characteristics of experienced pain.

The first category includes 10 groups of words that describe the sensory qualities of pain (Groups 1–10).The second category contains 5 groups of words referring to the emotional aspect of pain (Groups 11–15).The third category consists of 1 group of words that relate to the overall evaluation of pain (subjective experience) (Group 16).The fourth category (referred to as miscellaneous) is an additional category comprising 4 groups: 3 groups describing sensory pain qualities and 1 group referring to emotional qualities (Groups 17–19 and Group 20, respectively).

The questionnaire also includes a body diagram, on which the patient can indicate the location of pain.

The questionnaire defines the following parameters:Pain Rating Index—average values (PRI);Sum of average values from the sensory category (Groups 1–10) (S);Sum of average values from the affective category (Groups 11–15) (A);Average value from the evaluative category (Group 16) (E);Sum of average values from the miscellaneous sensory category (Groups 17–19) (M(S));Average value from the miscellaneous evaluative category (Group 20) (M(E));Sum of average values from the entire miscellaneous category (Groups 17–20) (M(O));Overall Pain Rating Index—sum of average values from Groups 1–20 (PRI(O)).

4.Beck Depression Inventory [15]The Beck Depression Inventory (BDI) is one of the most widely used self-report instruments for assessing the severity of depressive symptoms. It was originally developed by Aaron T. Beck and colleagues in 1961, and subsequent revisions (BDI-IA, BDI-II) were aligned with the prevailing diagnostic criteria (e.g., DSM). The scale consists of 21 items, each addressing a specific symptom of depression (e.g., depressed mood, feelings of guilt, sleep disturbances, loss of appetite). Each item is rated on a four-point scale (0–3), which reflects the intensity of the given symptom. The total score ranges from 0 to 63 points, with higher scores indicating greater severity of depressive symptomatology. Depending on the version and cultural adaptation, different cut-off scores are applied. In the Polish adaptation [16], the following thresholds are commonly used:-0–13 points—no depression or minimal symptoms,-14–19 points—mild episode,-20–28 points—moderate episode,-≥29 points—severe episode.

### 2.3. Statistical Analysis

The statistical analysis of qualitative data obtained from medical records describing the characteristics of study participants was presented as the absolute number of cases in each category (*n*) together with their relative proportion within the group (%). Quantitative data derived from the Visual Analogue Scale (VAS), the McGill Pain Questionnaire, and the Beck Depression Inventory (BDI) were analysed using statistical tests, including Student’s *t*-test, the Wilcoxon signed-rank test, the Shapiro–Wilk test, and the Chi-square test of independence, in addition to descriptive statistics. All analyses were conducted with a 5% margin of error, and statistical significance was set at *p* < 0.05, indicating meaningful differences between the study groups. Statistical computations were performed using the STATISTICA software package version 13 (StatSoft, Tulsa, OK, USA).

## 3. Results

### 3.1. Characteristics of the Study Group

The analysis began with a discussion of the sociodemographic data of the participants collected from the records. The mean age of the study group was 35.84 years (SD 11.9 years). The duration of the illness ranged from 1 to 11 years (SD 3.9 years). All patients in the study group were receiving pharmacological treatment (100%). Among the study group, 43 patients (71.67%) were women who already had children, while 17 (28.33%) were childless (Table 1).

The analysis of the data indicates that the majority of the women surveyed live in cities with populations exceeding 50,000 inhabitants. Twenty percent of the women reside in rural areas. Most of them lived in cities with populations over 50,000, i.e., 29 women; 12 participants lived in rural areas, and 19 lived in cities with populations up to 10,000. The analysis of the data regarding the education level of the study group, shows that 65% of the patients had higher education, 18.33% had secondary education, and 16.67% had vocational education.

### 3.2. Assessment of Pain Intensity

In this study, pain was assessed using the Visual Analog Scale (VAS) and the Melzack Pain Questionnaire, referring to pain experienced during menses and throughout the menstrual cycle. In the VAS assessment, patients indicated the intensity of pain by marking a point on the scale, where the left end represented no pain, and the right end indicated the highest pain intensity. In the Melzack Pain Questionnaire, patients selected one word from each group describing their pain sensations.

During menses, 8 patients reported mild pain, 17 patients reported moderate pain, 23 patients reported severe pain, and 12 patients reported excruciating pain. None of the patients in the study group indicated the absence of pain symptoms. During the menstrual cycle, 16 patients experienced mild pain, 27 patients moderate pain, 13 patients severe pain, and 4 patients excruciating pain. Again, none of the patients reported the absence of pain symptoms. Chi-square test of independence showed a significant association between menstrual phase and pain intensity, χ^2^(3) = 11.72, *p* = 0.008. The effect size was moderate (Cramér’s V = 0.31). Analyses were based on 60 participants, each assessed in two phases (*n* = 120 observations in total) (Table 2).

Figure 2 presents an analysis of the results regarding the percentage distribution of pain intensity as assessed using the VAS. During menses, 13% of patients reported mild pain, 28.33% reported moderate pain, 38.33% reported severe pain, and 20% reported excruciating pain. None of the patients in the study group reported an absence of pain symptoms. During the rest of the cycle, 26.67% of patients experienced mild pain, 45% reported moderate pain, 22% reported severe pain, and 6.67% experienced excruciating pain. Again, none of the patients in the study group reported an absence of pain symptoms.

Comparative analysis of pain intensity (VAS, 0–12) revealed that mean pain scores were significantly higher during menses (M = 7.23, SD = 3.13) compared to the non-menstrual phase of the cycle (M = 4.55, SD = 2.27). The difference was statistically significant in both the paired-samples *t*-test (t(59) = 11.61, *p* < 0.001) and the Wilcoxon signed-rank test (W = 31, *p* < 0.001). The effect size was very large (Cohen’s d = 1.50). Analysis of transitions between pain categories confirmed that, for the majority of participants, pain intensity decreased outside menstruation, most frequently shifting from the “severe” to the “moderate” category. Results of the analysis are provided in Table 3.

The calculation of pain intensity in the individual categories of the Melzack questionnaire was the next stage of the statistical analysis. The pain intensity results for the study group are summarized in Table 4. The following results demonstrate statistically significant differences in the perception of pain intensity during menses and throughout the menstrual cycle.

The present analysis involved a qualitative evaluation of pain sensations across multiple categories, alongside an assessment of accompanying symptoms and their characteristics. In the sensory category, during menses, patients most frequently described their pain using the terms throbbing (39), stabbing (33), pricking (31), stabbing (17), digging (28), cutting (18), pressing (15), crushing (27), tearing (20), ripping (31), burning (29), stinging (34), dull (49), sawing (26), and splitting (20). Descriptors such as flickering, twitching, blurred, and muffled were not employed. During the menstrual cycle, the most common descriptors were penetrating (49), boring (27), cutting (44), pressing (35), stretching (49), hot (47), tingling (36), muffled (22), and tightening (51). Conversely, descriptors including twitching, pounding, thumping, stabbing, crushing, tearing, burning, itching, breaking, and splitting were not selected. In the emotional category, menstruation was most often characterized as exhausting (53), nauseating (50), terrible (26), frightening (33), tormenting (37), and overwhelming (55). During the menstrual cycle, patients most frequently reported their pain as tiring (35), exhausting (25), nauseating (51), incredible (52), annoying (34), tormenting (26), blinding (42), and overwhelming (18). The descriptors frightening, maltreating, cruel, and deadly were not chosen. With regard to the evaluative category, pain during menses was predominantly described as awful (26) and unbearable (23), whereas the descriptor unpleasant was not selected. By contrast, during the menstrual cycle, the most frequent descriptors were unpleasant (18), disagreeable (25), and bothersome (14), while unbearable was not employed. In the diverse-sensory category, menstruation was most frequently associated with radiating (31), penetrating (12), and pulling (21). The descriptors cold, chilly, and icy were not used. During the menstrual cycle, the terms most commonly selected were expanding (25), radiating (22), tightening (26), dragging (22), and numbing (12), while penetrating, squeezing, pulling, cold, chilly, and icy were not employed. In the diverse-evaluative category, patients most often described their pain during menstruation as disgusting (24) and nasty (16). In contrast, during the menstrual cycle, the most frequent descriptors were annoying (37) and nasty (10), with disgusting and torturous not being chosen.

The analysis of accompanying symptoms demonstrated that gastrointestinal complaints were highly prevalent during menstruation, including constipation (23%), diarrhea (25%), and vomiting (14%). Furthermore, headaches and dizziness were reported by 15% and 17% of respondents, respectively, while sleepiness occurred in 6% of patients, which appeared to be associated with pain-related sleep disturbances. During the menstrual cycle, the predominant symptoms were headaches (25%), dizziness (17%), and sleepiness (23%). Gastrointestinal symptoms were reported by 35% overall, with constipation (18%), diarrhea (11%), and vomiting (6%). Regarding the nature of pain, the findings indicate that during menses the vast majority of patients (87%) experienced continuous pain, whereas 10% reported intermittent pain and 3% transient pain. In contrast, during the menstrual cycle, intermittent pain predominated (61%), followed by transient (27%) and continuous (12%) pain. Sleep disturbances also revealed marked differences between phases. During menstrual period, 51% of participants suffered from insomnia, 37% described their sleep as restless, and only 12% reported good sleep quality. By comparison, during the cycle, restless sleep was reported by 39%, good sleep quality by 38%, and insomnia by 23%. Finally, activity levels were substantially reduced during menstruation, with 59% of respondents reporting limited activity, 24% slight activity, 14% complete inactivity, and only 3% maintaining full activity. Across the menstrual cycle, however, 46% of patients reported limited activity, 31% maintained full activity, 16% slight activity, and 7% reported no activity at all (Table 5).

The Beck Depression Inventory was administered to the study participants. Each patient completed the questionnaire twice: once during menstruation and once during a non-menstrual phase. Chronic pain experienced by women with endometriosis was found to substantially impair the performance of basic daily activities and exert a negative impact on occupational functioning, family life, and social interactions. These effects manifested through limitations in physical capacity, disruptions of interpersonal relationships within the family, and difficulties in fulfilling physiological needs, frequently contributing to the development of depressive states. The collected data were scored according to the questionnaire scale and are presented in the table below. During menstrual period, the distribution of depressive disorders was as follows: 6 patients showed no symptoms of depression, 15 patients exhibited mild depressive symptoms, 15 patients demonstrated symptoms of moderate depression, and 24 patients presented with symptoms of severe depression. Across the menstrual cycle, the distribution of depressive symptoms was as follows: 12 patients reported no symptoms of depression, 18 patients exhibited mild depressive symptoms, 12 patients demonstrated symptoms of moderate depression, and 18 patients presented with symptoms of severe depression. Results of the analysis are provided in Table 6.

During menses, 40% of patients reported symptoms at the level of very severe depression, while 25% of the group exhibited signs of moderately severe and mild depression, respectively. No depressive symptoms were found in 10% of the respondents. Throughout the cycle, 30% of patients experienced symptoms of very severe depression, 30% showed signs of mild depression, and 20% of the studied group displayed moderately severe depressive symptoms or no symptoms at all. Table 7 presents the descriptive statistics for BDI scores obtained during the menstrual cycle and menstruation, as well as the calculated differences between the two conditions.

The mean BDI score during menstruation was clearly higher than during the cycle (30.1 vs. 22.2). An analysis of individual differences indicated that in the vast majority of participants (85%, N = 51), BDI scores were higher during menses compared to the cycle. Lower scores during menstruation were observed in eight women (13.3%), while only one participant (1.7%) showed no change. To compare BDI scores across the two conditions, both the paired-samples *t*-test and the non-parametric Wilcoxon signed-rank test were employed (Table 8).

The Shapiro–Wilk test was conducted to verify the assumption of normality for the difference scores. The result (W = 0.963, *p* = 0.065) indicated no significant deviation from normality, justifying the use of a parametric test. To ensure robustness, the non-parametric Wilcoxon test was also performed, which confirmed the same pattern of results. The mean BDI score during menstruation (M = 30.12, SD = 14.48) was significantly higher than during the cycle (M = 22.15, SD = 11.69). The mean difference was 7.97 points, 95% CI [5.3; 10.6]. Results of the paired *t*-test confirmed this difference, t(59) = 6.02, *p* < 0.001, with a large effect size (Cohen’s d = 0.78). The Wilcoxon signed-rank test yielded consistent findings, W = 204.5, Z = −5.23, *p* < 0.001, with a large effect size (r = 0.68). Statistical tests consistently demonstrated that depressive symptoms, as measured by the BDI, were significantly more severe during menses compared to the cycle. The effect sizes (Cohen’s d = 0.78; r = 0.68) further indicate that this difference is not only statistically significant but also clinically meaningful, reflecting a substantial increase in depressive symptomatology during menses.

The results presented in Table 9 indicate that among the studied group of patients during menstruation and during the cycle, the absence of depressive symptoms did not correlate with any level of pain intensity. Mild depressive symptoms were reported by 57% of patients experiencing mild pain, 16% of those with moderate pain, and 10% of patients experiencing severe or unbearable pain. Symptoms of moderately severe depression were reported by 43% of patients with mild pain, 49% of those with moderate pain, and 47% of patients with severe or unbearable pain. Symptoms of very severe depression were not observed in patients with mild pain but were present in 35% of patients with moderate pain and 43% of those experiencing severe or unbearable pain. The research results indicating that, in the studied group of patients during the cycle, the absence of depressive symptoms did not correlate with any level of pain intensity. Symptoms of mild depression were reported by 80% of patients experiencing mild pain, 55% of those with moderate pain, and 45% of those suffering from severe or unbearable pain. Symptoms of moderate depression were observed in 20% of patients with mild pain, 45% with moderate pain, and 48% with severe or unbearable pain. Symptoms of very severe depression were not reported by patients with mild or moderate pain but were present in 7% of patients experiencing severe or unbearable pain

To examine whether the intensity of depressive symptoms varied as a function of pain severity, two separate contingency tables were constructed for the menstruation phase and the non-menstrual phase of the cycle. Patients were categorized according to their reported pain intensity (mild, moderate, severe/excruciating) and the level of depressive symptoms (mild, moderate, severe). For each phase, the distribution of observed frequencies was compared with the distribution expected under the null hypothesis of independence using the Pearson chi-square test of independence.

During menstruation, the association between pain intensity and depressive symptom severity was statistically significant, χ^2^(4) = 12.89, *p* = 0.012. Inspection of the cross-tabulation (Table 10) indicated that patients with moderate or excruciating pain were more likely than expected to report severe depressive symptoms, whereas those with mild pain were less likely to report such symptoms. By contrast, in the non-menstrual phase, no significant association was found between pain intensity and depressive symptom severity, χ^2^(4) = 6.66, *p* = 0.155, suggesting that the relationship between pain and mood disturbances is particularly pronounced during menses.

## 4. Discussion

Endometriosis constitutes a major global public health burden, affecting a substantial proportion of women of reproductive age. Pain—commonly misinterpreted by patients as physiological dysmenorrhea and thus normalized as an inherent element of the menstrual cycle—frequently represents the earliest clinical manifestation of a chronic and progressive disorder that significantly compromises functional capacity and health-related quality of life (HRQoL) [17]. Epidemiological estimates indicate that approximately 10% of women of reproductive age are affected [18,19,20]. Within this cohort, the prevalence of depressive symptomatology has been reported to range between 20% and 85%, underscoring the profound psychiatric comorbidity associated with the disease. Pain intensity and persistence are consistently regarded as principal indicators of disease trajectory and progression [17]. A bibliometric survey of PubMed-indexed literature revealed 185 publications within the past five years containing the terms “endometriosis,” “pain,” and “depression,” reflecting a rapid expansion of scientific inquiry yet simultaneously exposing significant knowledge deficits warranting further systematic investigation.

The nociceptive profile of endometriosis is predominantly chronic, with peak exacerbations in the perimenstrual period. Quantification via the Visual Analogue Scale (VAS) revealed that, during menstruation, pain was rated as mild in 8 patients, moderate in 17, severe in 23, and excruciating in 12. Outside of menstruation, the corresponding distribution was mild in 16 patients, moderate in 27, severe in 13, and excruciating in 4. Application of percentage distribution analysis, Chi-square testing, Student’s *t*-test, and the Wilcoxon signed-rank test demonstrated statistically significant differences across cycle phases. Complementary findings by Gouesbet et al. indicated that patients reported moderate-to-severe pain irrespective of disease stage or menstrual phase [21]. During menstruation, symptoms of mild depression were reported in 57% of patients experiencing mild pain, in 16% of those reporting moderate pain, and in 10% of those suffering from severe pain. Moderate depressive symptoms were observed in 43%, 49%, and 47% of participants in the respective groups, whereas severe depressive symptoms were documented in 0%, 35%, and 43% of cases, respectively. It is noteworthy that none of the patients were entirely free from depressive symptomatology. Outside the menstrual period, mild depressive symptoms were reported by 80% of women experiencing mild pain, 55% of those with moderate pain, and 45% of those with severe pain. Moderate depressive symptoms were present in 20%, 45%, and 48% of participants in the respective categories, while severe depressive symptoms were reported exclusively among women experiencing severe pain (7%). These findings are consistent with extant empirical evidence indicating a robust association between the presence of somatic disorders and the manifestation of depressive symptoms. In a population-based study, Bello et al. [22] demonstrated that the severity of menstrual pain exhibited the strongest association with the presence of mental health disorders, including depression. Concordant conclusions were reported by Papkour et al. [23] in a systematic review, which identified a significant correlation between dysmenorrhea and affective disturbances, including depression, anxiety, and stress. Furthermore, Gambadauro and colleagues [24] conducted a cross-sectional investigation among adolescent females, with a mean participant age of 14.1 years. Using the Beck Depression Inventory (BDI) and an author-designed questionnaire, the study found that 581 respondents-constituting 55.1% of the study population-reported experiencing painful menstruation. Subsequent analyses revealed that these individuals were more likely to report concomitant mental health complaints. Importantly, the authors emphasized a bidirectional association: the presence of psychological symptoms, such as depressive mood, significantly increased the likelihood of dysmenorrhea. Specifically, the prevalence of dysmenorrhea was approximately 29–34% higher among girls presenting with psychological symptomatology compared with their asymptomatic counterparts [24].

A retrospective analysis demonstrated that 40% of respondents experienced severe depressive symptoms during menstruation, with prevalence decreasing to 30% during the intermenstrual interval. Scores on the Beck Depression Inventory (BDI) were significantly lower during menstruation compared to other cycle phases. Moreover, Chi-square analysis confirmed a significant association between pain phenotype and the occurrence of depressive symptomatology. In a study by Škegro et al. [25] employing the DASS-21 instrument, depressive symptoms of varying severity were observed in 44.3% of patients with histopathologically verified endometriosis post-surgery [25]. A systematic review by Szypłowska et al. [20] reported prevalence estimates ranging from 9.8% to 98.5%, emphasizing the substantial psychosocial burden of the disease independent of the assessment methodology. In several of these investigations, the BDI was employed [20]. Furthermore, Mormile and Picone [26] advanced the hypothesis of a potential causal nexus between endometriosis and depression, mediated by immunological dysregulation. Specifically, the pathophysiology of endometriosis is characterized by elevated levels of myeloid-derived suppressor cells (MDSCs) and concomitant reduction in regulatory T cell (Treg) expression. These immune alterations may potentiate the development of depression via homologous immunopathogenic pathways [26].

Sleep disturbance constitutes an additional clinically salient domain of disease burden. During menstruation, 51% of patients reported insomnia, 37% described sleep as restless, and only 12% characterized sleep quality as satisfactory. In non-menstrual phases, 39% reported restless sleep, 38% satisfactory sleep, and 23% insomnia. Evidence from de Souza et al. [27] demonstrated that patients reporting moderate-to-severe pain were significantly more likely to experience insomnia compared with those reporting mild or absent pain. Duration of pain emerged as an independent risk factor: women experiencing moderate-to-severe pain reported threefold longer symptom duration relative to those with mild or absent pain, with the risk of insomnia doubling per decade of disease chronicity. Pain catastrophizing further exacerbates sleep disturbances, particularly in the absence of efficacious therapeutic interventions [27]. Corroborative findings by Ishikura et al. [28] revealed that sleep impairment may act as a pro-inflammatory driver, thereby amplifying disease activity and pain perception in endometriosis [28].

This study has several limitations related to its retrospective observational design. The analysis relied on pre-existing medical records, which may have been incomplete or inconsistent, introducing a risk of bias. The representativeness of the study population was limited by the availability of data, which constrains the generalizability of the findings. Finally, although appropriate statistical methods were applied, potential confounding factors could not be fully controlled.

## 5. Conclusions

Menstrual pain was found to be significantly more intense than pain experienced during non-menstrual phases of the cycle. Statistical analyses demonstrated that mean pain intensity, as measured with the Visual Analogue Scale (VAS), was markedly higher during menstruation compared to the non-menstrual phase. This difference was associated with a large effect size, indicating a clinically meaningful increase in pain severity during menstruation.The qualitative characteristics of pain varied according to the phase of the cycle. Analysis of the McGill Pain Questionnaire revealed that the menstrual phase was dominated by descriptions of continuous, severe, and emotionally burdensome pain, whereas the non-menstrual phase was more frequently associated with transient and less psychologically taxing pain descriptors.Accompanying symptoms were more pronounced during menstruation. Gastrointestinal disturbances (including diarrhea, constipation, and vomiting) occurred more frequently in the menstrual phase, alongside a higher prevalence of sleep disturbances and reduced physical activity. Conversely, outside menstruation, patients more commonly reported headaches, drowsiness, and improved sleep quality.Depressive symptomatology was significantly exacerbated during menstruation. Beck Depression Inventory (BDI) scores indicated a higher severity of depressive symptoms during the menstrual phase compared to the non-menstrual phase. This difference reached statistical significance and was associated with a large effect size, suggesting a clinically relevant aggravation of mood disturbances concurrent with menstrual pain.The relationship between pain intensity and depressive symptomatology was particularly evident during menstruation. A chi-square test demonstrated that during the menstrual phase, pain severity was significantly associated with the severity of depressive symptoms (χ^2^(4) = 12.89, *p* = 0.012). Patients experiencing moderate to severe/excruciating pain were more likely than expected to report severe depressive symptoms. In contrast, this association was not statistically significant during the non-menstrual phase (χ^2^(4) = 6.66, *p* = 0.155).

## 6. Limitations of the Study

The retrospective observational design inherently involves certain methodological constraints that may affect the interpretation and generalizability of results. The most important limitations identified in this study include the following:-Population and sample selection.

The study population was determined by the availability of data from existing sources, which may have limited the representativeness of the sample. This constraint introduces a potential risk of selection bias. To minimize this risk, available datasets were carefully reviewed, and potential sources of bias were critically considered.

-Data collection.

Retrospective research relies on historical data, which may be incomplete or inconsistent. A systematic and rigorous approach was applied to identify and extract all relevant information from medical records to ensure maximal completeness and accuracy of the dataset.

-Generalizability.

Results from retrospective studies may not be directly generalizable to other populations or time periods, as they are derived from specific cohorts and historical contexts. To mitigate this limitation, a representative sample was selected, and appropriate statistical methods were applied to reduce potential biases.

-Reliability and validity of data.

The accuracy of retrospective analyses depends on the quality of source material. Inconsistencies or missing data may undermine reliability and validity. To minimize this risk, data were collected from consistent and verified medical sources whenever possible.

-Data analysis and interpretation.

Statistical analysis in retrospective studies must account for potential confounding factors and the historical context of the data. Methods such as regression analysis, analysis of variance, and trend analysis were considered appropriate. Interpretation of results was performed with awareness of possible limitations due to the retrospective nature of the study.

## Figures and Tables

**Figure 1 diseases-13-00291-f001:**
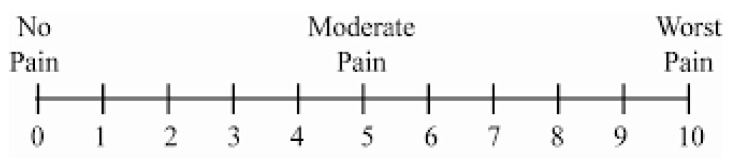
Visual Analogue Scale.

**Figure 2 diseases-13-00291-f002:**
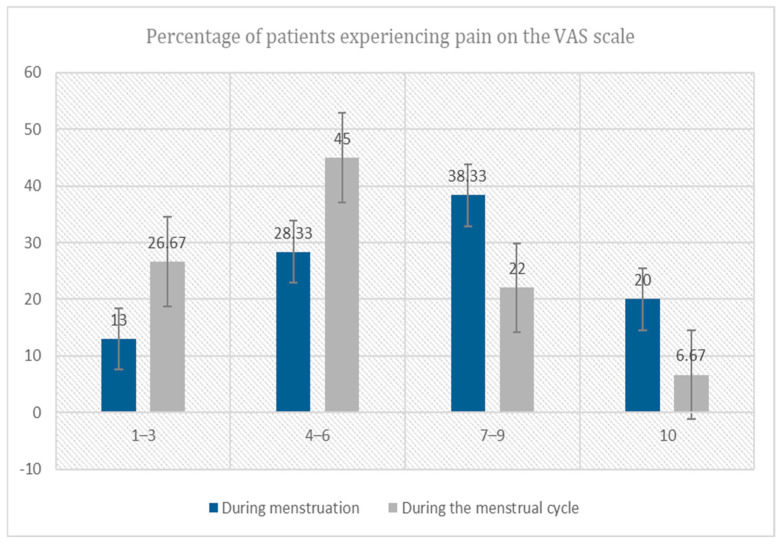
Percentage distribution of pain intensity experienced by the study group patients assessed using the VAS during menstruation and the rest of the cycle. χ^2^(3) = 11.72, df = 3, *p* = 0.008, Cramér’s V = 0.31.

**Table 1 diseases-13-00291-t001:** Characteristics of the patient population in the study group.

Study Group (*n* = 60)	
Age (years) Mean (SD)	35.84 (11.19)
Duration of illness (years) Mean (SD)	1–11 (3.9)
Patients of reproductive age	60 (100%)
Treatment methods: Pharmacological	60 (100%)
Patients with children	43 (71.67%)
Childless patients	17 (28.33%)

**Table 2 diseases-13-00291-t002:** Pain intensity reported by patients in the study group using the VAS during menstruation and the menstrual cycle.

Pain Intensity VAS	None (0)	Mild (1 2 3)	Moderate (4 5 6)	Severe (7 8 9)	Excruciating (10)
During menstruation	0	8 (13.3%)	17 (28.3%)	23 (38.3%)	12 (20.0%)
	0	16 (26.7%)	27 (45.0%)	13 (21.7%)	4 (6.7%)

χ^2^(3) = 11.72, df = 3, *p* = 0.008, Cramér’s V = 0.31.

**Table 3 diseases-13-00291-t003:** Descriptive statistics and results of comparative tests for pain intensity (VAS).

Condition/Test	M	SD	Median	Min	Max	Statistic	df/N	*p*	Mean Difference	95% CI of Difference	Cohen’s d
During menstruation	7.23	3.13	7.0	2	12	-	-	-	-	-	-
During cycle	4.55	2.27	5.0	1	9	-	-	-	-	-	-
Paired-samples *t*-test	-	-	-	-	-	11.614	59	<0.001	2.68	[2.22; 3.15]	1.499
Wilcoxon signed-rank	-	-	-	-	-	31.000	60	<0.001	2.68	-	-

**Table 4 diseases-13-00291-t004:** Analysis of pain intensity in individual categories of the Melzack. Questionnaire in the study group of patients during menstruation and the cycle.

Category Name	During Menstruation		During the Cycle	
	M	SD	M	SD
Sensory	25.28	2.79	10.46	5.01
Emotional	16.14	1.51	8.33	4.01
Evaluative (subjective)	3.49	0.82	2.62	0.54
Miscellaneous Evaluative	8.23	0.92	5.36	1.91
Varied evaluative	7.10	0.62	3.70	1.16
Miscellaneous	15.01	1.52	9.05	3.11
Overall pain score	61.20	5.81	31.03	12.57

**Table 5 diseases-13-00291-t005:** Comprehensive summary of symptoms, pain characteristics, sleep disturbances, and activity levels among patients during menstruation and across the menstrual cycle.

Section	Category	During Menstruation (%)	During the Cycle (%)
Accompanying symptoms			
	Diarrhea	25	11
	Vomiting	14	6
	Constipation	23	18
	Headaches	17	25
	Dizziness	15	17
	Sleepiness	6	23
Nature of pain			
	Continuous pain	87	12
	Intermittent pain	10	61
	Brief pain	3	27
Sleep disturbances			
	Good sleep	12	38
	Restless sleep	37	39
	Insomnia	51	23
Activity assessment			
	Full activity	3	31
	Limited activity	59	46
	Minimal activity	24	16
	No activity	14	7

**Table 6 diseases-13-00291-t006:** Assessment of the severity of depressive disorders in the studied group of patients during menstruation and throughout the menstrual cycle, conducted using the Beck Depression Inventory.

Severity of Depressive Disorders	No Depression	Mild Depression	Moderate Depression	Severe Depression
Beck’s scale	0–13	14–19	20–28	29–63
Menstruation (*n* = 60)	6 (10%)	15 (25%)	15 (25%)	24 (40%)
Cycle (*n* = 60)	12 (20%)	18 (30%)	12 (20%)	18 (30%)

**Table 7 diseases-13-00291-t007:** Descriptive statistics for BDI scores.

Measure	Cycle (*n* = 60)	Menstruation (*n* = 60)	Difference (*n* = 60)
Mean	22.15	30.12	7.97
Median	19.5	27.0	5.0
Minimum	6	10	−14
Maximum	55	60	33
Std. Deviation	11.69	14.48	10.26
Q1 (25%)	14.0	18.0	1.75
Q3 (75%)	29.0	46.0	13.0

**Table 8 diseases-13-00291-t008:** Results of comparative analyses of pain intensity during menstruation and the non-menstrual phase (paired *t*-test and Wilcoxon signed-rank test).

Test	Statistic	*p*-Value	Conclusion
Paired *t*-test	t(59) = 6.02	*p* < 0.001	Significant
Wilcoxon signed-rank	W = 204.5, Z = −5.23	*p* < 0.001	Significant

**Table 9 diseases-13-00291-t009:** Depressive symptoms by pain intensity during menstruation vs. the non-menstrual phase.

Pain Intensity	Depressive Symptoms	Menstruation (%)	Cycle (%)
Mild pain	No depression	0%	0%
	Mild depression	57%	80%
	Moderate depression	43%	20%
	Severe depression	0%	0%
Moderate pain	No depression	0%	0%
	Mild depression	16%	55%
	Moderate depression	49%	45%
	Severe depression	35%	0%
Severe/excruciating pain	No depression	0%	0%
	Mild depression	10%	45%
	Moderate depression	47%	48%
	Severe depression	43%	7%

**Table 10 diseases-13-00291-t010:** Distribution of depressive symptom severity (mild, moderate, severe) across pain intensity categories (mild, moderate, severe/excruciating) during menstruation and in the non-menstrual phase.

Phase of Cycle	Pain Intensity	Mild Depression (obs/exp)	Moderate Depression (obs/exp)	Severe Depression (obs/exp)	Total
Menstruation	Mild	8/3.5	6/6.5	0/4.0	14
	Moderate	5/7.7	15/14.1	11/9.2	31
	Excruciating	2/3.8	7/7.4	6/4.8	15
Total (mens.)		15	28	17	60
Cycle	Mild	12/8.8	3/5.9	0/0.3	15
	Moderate	16/17.0	13/11.3	0/0.6	29
	Excruciating	7/9.2	8/6.8	1/0.4	16
Total (cycle)		35	24	1	60

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
