# Peer review of "Assessment of the Impact of Chronic Pain on the Prevalence of Depressive Disorders in Patients with Endometriosis"

_diseases, 2025, doi:10.3390/diseases13090291_

Round 1
Reviewer 1 Report
Comments and Suggestions for Authors
This manuscript titled ‘Assessment of the Impact of Chronic Pain on the Prevalence of Depressive Disorders in Patients with Endometriosis’ by Rysiak et al., aims to address the association between chronic pain in endometriosis and depressive disorders, and provide clinical insights into the integration of psychological care into endometriosis management.
Overall, the study is designed and performed in a scientific manner. The manuscript is well-structured and well-written. I only have several specific questions.
- Please provide statistical analysis for all the figures.
- The authors mentioned that they used Mann-Whitney U tests, but it’s unclear how these were applied to multiple comparisons. The authors need to clarify correction methods for multiple testing (e.g., Bonferroni, Benjamini, or FDR).
- I noticed that the Beck Depression Inventory (BDI) categories were used in this study, but the thresholds (0–11, 12–26, etc.) differ slightly from standard BDI cut-offs. The justification is missing. The authors also need to discuss whether scores reflect clinical diagnosis or screening results.
- I suggest that the authors provide a paragraph to discuss the limitations of the current study. For example, how ‘single-center’ and ‘Polish cohort’ could potentially affect the generalization of the conclusion? How would the concurrent pharmacological treatment affect pain/depression scores?
- Some sentences are long and could be made more concise. Minor grammatical corrections needed (e.g., “menstruation and throughout the menstrual cycle” is repetitive).
- Figures 3–6 could be merged or simplified to improve readability.
- Lastly, while informed consent is noted in the manuscript. Please specify the ethical approval reference number and approving body.
Author Response
1. Summary |
|
|
Thank you very much for taking the time to review this manuscript. Please find the detailed responses below and the corresponding revisions, corrections highlighted, in track changes in the re-submitted files.
|
||
2. Questions for General Evaluation |
Reviewer’s Evaluation |
Response and Revisions |
Does the introduction provide sufficient background and include all relevant references? |
Yes
|
|
Are all the cited references relevant to the research? |
Yes |
|
Is the research design appropriate? |
Yes |
|
Are the methods adequately described? |
Can be improved |
We would like to thank you for this valuable comment. The suggested revisions have been incorporated into the manuscript. |
Are the results clearly presented? |
Can be improved |
We would like to thank you for this valuable comment. The suggested revisions have been incorporated into the manuscript. |
Are the conclusions supported by the results? |
Can be improved |
We would like to thank you for this valuable comment. The suggested revisions have been incorporated into the manuscript.
|
3. Point-by-point response to Comments and Suggestions for Authors |
||
Comments 1: Please provide statistical analysis for all the figures. |
||
Response 1: We thank the Reviewer for this valuable comment. In the revised manuscript, we have provided detailed information on the statistical analyses conducted for the data presented in the tables and figures. Specifically, we have added descriptions of the statistical tests applied and the corresponding p-values, where appropriate. These details have now been included in the legends for greater clarity. For data presenting descriptive or schematic information, statistical analysis was not applicable, and this has been explicitly indicated in the respective descriptions.
|
||
Comments 2: The authors mentioned that they used Mann-Whitney U tests, but it’s unclear how these were applied to multiple comparisons. The authors need to clarify correction methods for multiple testing (e.g., Bonferroni, Benjamini, or FDR). Response 2: We agree. Accordingly, we have revised and updated the manuscript to address this point. The collected data were reanalyzed and new statistical analyses were performed. For each table and figure, the applied statistical tests have been specified. This aspect has been further elaborated in the Statistical Analysis section. The data were analyzed using statistical tests, including the Student’s t-test, the Wilcoxon signed-rank test, the Shapiro–Wilk test, and the chi-square test of independence, in addition to descriptive statistics.
|
||
Comments 3: I noticed that the Beck Depression Inventory (BDI) categories were used in this study, but the thresholds (0–11, 12–26, etc.) differ slightly from standard BDI cut-offs. The justification is missing. The authors also need to discuss whether scores reflect clinical diagnosis or screening results. |
||
Response 3: Agree. We have therefore made changes to indicate this correctly. Indeed, an error occurred. After the patients selected their responses, the total score was calculated as the sum of all items reflecting their self-reported psychophysical state during the week preceding the assessment. The interpretation of the results was subsequently conducted in accordance with the following scale: 0-11 points- absence of depression or a transient lowering of mood, 12-18 points- mild depression, 19-26 points- moderate depression, 27-49 points- severe depression, 50-63 points- very severe depression.
Furthermore, in line with the reviewer’s suggestion, we have emphasized in the revised manuscript that the BDI is a screening instrument, and its scores reflect the severity of self-reported depressive symptoms rather than serving as a basis for a clinical diagnosis of depression. We have also added clarification in the “Study Limitations” section, noting that the use of the BDI allows only for an estimation of the risk and severity of depressive symptoms, but does not permit their formal clinical diagnosis.
Comments 4: I suggest that the authors provide a paragraph to discuss the limitations of the current study. For example, how ‘single-center’ and ‘Polish cohort’ could potentially affect the generalization of the conclusion? How would the concurrent pharmacological treatment affect pain/depression scores? Response 4: We thank the Reviewer for this valuable comment. A retrospective analysis was conducted on data derived from the medical records of sixty female patients treated at the Tomaszewski Medical Center in Bialystok between 2023 and 2024. The study included women of reproductive age diagnosed with endometriosis-related chronic pain, without concomitant autoimmune diseases or infertility, and managed conservatively with hormonal pharmacotherapy. The analysis was based on the transcription and subsequent interpretation of data collected during routine clinical visits, with specific attention to the following variables: · age, · socio-economic and social functioning, · co-occurrence of symptoms and complaints negatively affecting patients’ daily functioning, · severity and characteristics of pain reported by the patients, · depressive symptoms.
Retrospective studies are based on the evaluation of data that have already been collected, in this case, during medical consultations and documented in patient records, in order to examine relationships between risk factors and clinical outcomes. The retrospective assessment of medical records focused on a defined patient cohort from a single medical center within a specific time frame, analyzed retrospectively in relation to the time of the study. All collected data were transferred into a Microsoft Excel spreadsheet, structured into cells to create an organized dataset. The accuracy of data entry was verified using the “data validation” function, which applied restrictions on input values to ensure consistency and correctness of the entered variables.
Comments 5: Some sentences are long and could be made more concise. Minor grammatical corrections needed (e.g., “menstruation and throughout the menstrual cycle” is repetitive). Response 5: Agree. Thank you for this helpful comment. We have revised the manuscript to improve clarity and readability by shortening long sentences and correcting minor grammatical issues. The repetitive phrasing has been corrected accordingly.
Comments 6: Figures 3–6 could be merged or simplified to improve readability. Response 6: We thank the Reviewer for this suggestion. We have carefully reconsidered Figures 3–6 and agree that their presentation could be improved. Accordingly, we have simplified the layout and formatting to enhance readability. The data have been presented in a consolidated form within a table. We hope that this modification will improve the clarity of the data and have a positive impact on their interpretation.
Comments 7: Lastly, while informed consent is noted in the manuscript. Please specify the ethical approval reference number and approving body. Response 7: We thank the Reviewer for this important comment. We would like to emphasize that this study was retrospective in nature and was based solely on anonymized patient questionnaires extracted from medical records. In accordance with the applicable national regulations (Act on the Professions of Physician and Dentist, Journal of Laws 1997 No. 28, item 152, as amended) and institutional guidelines, retrospective analyses using anonymized medical data, without any intervention in patient treatment, do not require approval from a bioethics committee. Nevertheless, we have explicitly clarified this point in the revised manuscript to make it clear that ethical approval was not required for this type of study.
|
||
4. Additional clarifications |
||
We would like to thank the Reviewer for their constructive feedback, which has significantly improved the clarity and quality of our manuscript. All comments have been carefully addressed, and the corresponding changes have been incorporated into the revised version. We have also re-checked the manuscript to ensure consistency of terminology, accuracy of references, and overall readability. We trust that the revised version meets the expectations and hope that it will now be suitable for publication. |
Reviewer 2 Report
Comments and Suggestions for Authors
This is an interesting study. However, as an example, “pain” and “depression” co-exist in the title of more than 2.500 articles on pubmed. Thus, it seems that your results were roughly “anticipated”. You should further discuss this co-existence in the Discussion.
Author Response
1. Summary |
|
|
Thank you very much for taking the time to review this manuscript. Please find the detailed responses below and the corresponding revisions, corrections highlighted, in track changes in the re-submitted files.
|
||
2. Questions for General Evaluation |
Reviewer’s Evaluation |
Response and Revisions |
Does the introduction provide sufficient background and include all relevant references? |
Can be improved
|
We would like to thank you for this valuable comment. The suggested revisions have been incorporated into the manuscript.
|
Is the research design appropriate? |
Can be improved
|
We would like to thank you for this valuable comment. The suggested revisions have been incorporated into the manuscript. |
Are the methods adequately described? Are the results clearly presented?
Are the conclusions supported by the results?
Are all figures and tables clear and well-presented? |
Yes
Can be improved
Must be improved
Yes |
We would like to thank you for this valuable comment. The suggested revisions have been incorporated into the manuscript. We thank the Reviewer for this important comment. In response, we have carefully revised and expanded the manuscript to ensure a clearer, more detailed, and comprehensive presentation of the study. |
3. Point-by-point response to Comments and Suggestions for Authors |
||
Comments 1: This is an interesting study. However, as an example, “pain” and “depression” co-exist in the title of more than 2.500 articles on pubmed. Thus, it seems that your results were roughly “anticipated”. You should further discuss this co-existence in the Discussion. |
||
Response 1: We thank the Reviewer for this valuable comment. We agree that the co-existence of pain and depression has been widely reported in the literature, as demonstrated by the large number of PubMed-indexed articles addressing these phenomena. In response to this comment, we have expanded the Discussion section to more explicitly situate our findings within this broader body of evidence. In the revised manuscript, we now emphasize that although the association between pain and depression is well documented, relatively few studies have specifically examined this relationship in the context of endometriosis, with a focus on the intensity of menstrual pain and its correlation with depressive symptoms. Our study contributes to this literature by providing empirical evidence based on validated psychometric instruments (VAS, McGill Pain Questionnaire, Beck Depression Inventory) in a defined clinical sample. Furthermore, we highlight in the Discussion that our results are consistent with previous research showing a strong link between chronic pain and mood disturbances, but also underline the unique clinical relevance of addressing this relationship in women with endometriosis. We argue that recognizing this association may have implications for more comprehensive patient care, where both somatic and psychological dimensions are considered. We believe that these revisions adequately address the Reviewer’s concern by situating our work within the existing literature while clarifying the novel contribution of our study.
|
||
4. Additional clarifications |
||
We would like to thank the Reviewer for their constructive feedback, which has significantly improved the clarity and quality of our manuscript. All comments have been carefully addressed, and the corresponding changes have been incorporated into the revised version. We have also re-checked the manuscript to ensure consistency of terminology, accuracy of references, and overall readability. We trust that the revised version meets the expectations and hope that it will now be suitable for publication. |
Round 2
Reviewer 1 Report
Comments and Suggestions for Authors
The manuscript was improved. I don't have further comments.